# Out-of-equilibrium Eigenstate Thermalization Hypothesis

Laura Foini,[1, *] Anatoly Dymarsky,[2, §] and Silvia Pappalardi[3, †]

[1]*IPhT, CNRS, CEA, Université Paris Saclay, 91191 Gif-sur-Yvette, France*
[2]*Department of Physics and Astronomy, University of Kentucky, Lexington, KY 40503 USA*
[3]*Institut für Theoretische Physik, Universität zu Köln, Zülpicher Straße 77, 50937 Köln, Germany*
(Dated: July 4, 2024)

Understanding how out-of-equilibrium states thermalize under quantum unitary dynamics is an important problem in many-body physics. In this work, we propose a statistical ansatz for the matrix elements of non-equilibrium initial states in the energy eigenbasis. The approach is inspired by the Eigenstate Thermalisation Hypothesis (ETH) but the proposed ansatz exhibits different scaling. Importantly, exponentially small cross-correlations between the observable and the initial state matrix elements determine relaxation dynamics toward equilibrium. We numerically verify scaling and cross-correlation, point out the emergent universality of the high-frequency behavior, and outline possible generalizations.

**Introduction** - Over the past decades, the unitary evolution of nonequilibrium states, including post-quench dynamics, has been a prominent subject in the field of quantum dynamics. The mechanism for thermalization is now well understood via the Eigenstate Thermalization Hypothesis (ETH) [1–4]. The latter is a statistical ansatz for the matrix elements of physical observables $\hat{A}$ is the energy eigenbasis $\hat{H}|E_i\rangle = E_i|E_i\rangle$:

$$A_{ij} = \mathcal{A}(E^+)\delta_{ij} + e^{-S(E^+)/2}f_A(E^+,\omega_{ij})R_{ij} , \quad (1)$$

with $E^+ = (E_i + E_j)/2$, $\omega_{ij} = E_i - E_j$ being the average energy and frequency, $S(E)$ is thermodynamic entropy, and $R_{ij}$ is a pseudorandom variable, such that $\overline{R_{ij}} = 0$ and $\overline{R_{ij}R_{ji}} = 1$. Finally, $\mathcal{A}(E)$ and $f_A(E,\omega)$ are smooth functions of their arguments. This ansatz has proved to be extremely successful in describing the equilibrium dynamics [4, 5] of physical local Hamiltonians, as was shown by extensive numerical calculations [6–14]. Recently, the study of correlations between matrix elements [15] has led to novel developments beyond the standard framework, connecting ETH with Free Probability theory [15–18], random matrix universality [19–25], conformal field theories [26] and motivating the study of energy eigenvectors statistics [27–34].

One of the central questions is how to extend the ETH framework to describe *non-equilibrium* dynamics [2, 35–41]. In this work, we propose a statistical ansatz for the matrix elements of the projector on the initial out-of-equilibrium state $\Psi = |\psi\rangle\langle\psi|$ written in the eigenbasis of the Hamiltonian. Notably, the non-equilibrium dynamics are encoded in the *correlations between the initial state and the observable's off-diagonal matrix elements*, which we describe in our framework. After introducing the ansatz and verifying its consistency, we discuss its implications for the relaxation dynamics towards equilibrium and numerically verify it in a non-integrable one-dimensional spin chain.

**Set up** - The dynamics of a local observable can be written in the eigenbasis of the Hamiltonian as

$$\langle\psi|\hat{A}(t)|\psi\rangle = \sum_{ij} c_i c_j^* A_{ij} e^{i(E_i - E_j)t} . \quad (2)$$

with $c_i = \langle\psi|E_i\rangle$. The original ETH (1) is designed to describe the stationary equilibrium point. In the absence of degeneracies, the expectation value of $A$ eventually attains a stationary value

$$\sum_i |c_i|^2 A_{ii} = \langle\hat{A}\rangle_{\text{diag}} , \quad (3)$$

which can be described by standard statistical mechanics. Namely, one introduces the diagonal ensemble $\hat{\rho}_{\text{diag}} = \sum_i |c_i|^2 |E_i\rangle\langle E_i|$ such that $\langle\hat{A}\rangle_{\text{diag}} = \text{Tr}\left(\hat{A}\,\hat{\rho}_{\text{diag}}\right)$ [3, 42]. In this work, we consider pure initial states with extensive mean energy and sub-extensive energy fluctuations in the number of degrees of freedom $N$

$$\begin{aligned} \langle\psi|\hat{H}|\psi\rangle &= E_0 \simeq e_0 N, \\ \sqrt{\langle\psi|(\hat{H} - E_0)^2|\psi\rangle} &\simeq \delta_{e_0} N^a, \qquad a < 1. \end{aligned} \quad (4)$$

For such initial states, the stationary value of $\langle\hat{A}(t)\rangle$ is given by the microcanonical expectation, that, combined with ETH implies thermalization, i.e. $\langle\hat{A}\rangle_{\text{diag}} \simeq \mathcal{A}(E_0)$ [2, 3]. As a main example, we consider the case $a = 1/2$, satisfied if one performs a global quench, which also characterizes equilibrium ensembles. Nonetheless, we will discuss the validity of our ansatz also for other initial states (see the Discussions).

The fundamental object that we want to characterize is the projector on the initial state written in the basis of the Hamiltonian

$$\Psi_{ij} = c_i c_j^* = \langle E_j|\psi\rangle\langle\psi|E_i\rangle . \quad (5)$$

We will treat it as a pseudorandom object, analogously to $A_{ij}$ in ETH. A crucial difference, in comparison with $\hat{A}$, is that this operator is of rank one and that

each off-diagonal matrix element is the product of two pseudo-random numbers. This will radically change the scaling in the proposed ansatz. Crucially, to capture the out-of-equilibrium dynamics, we will assume that correlations exist between $\Psi$ and $\hat{A}$, when expressed in the energy eigenbasis.

**Ansatz** - We introduce an ansatz for the matrix $\Psi$,

$$\Psi_{ij} \simeq \frac{e^{-\Phi(E_i)}}{Z}\delta_{ij} + \frac{e^{-\frac{1}{2}(\Phi(E_i)+\Phi(E_j))}}{Z}\tilde{R}_{ij} , \quad (6)$$

where $\Phi(E)$ is a smooth function of energy and $Z = \sum_i e^{-\Phi(E_i)}$ the normalization, which defines the diagonal ensemble:

$$\overline{\Psi_{ii}} = |c_i|^2 = \frac{e^{-\Phi(E_i)}}{Z} . \quad (7)$$

In Eq.(6), $\tilde{R}_{ij}$ are pseudorandom variables with zero average and unit variance, i.e.

$$\overline{\tilde{R}_{ij}} = 0 , \quad \overline{\tilde{R}_{ij}^2} = 1, \quad \text{for } i \neq j.$$

Diagonal pseudo-random $\tilde{R}_{ii}$ also have zero average, but the particular value of their variance may depend on the symmetry class of $\hat{H}$ (e.g. GOE or GUE), as it is the case for $R_{ii}$ in standard ETH [19, 43]. Given that $\Psi_{ii}$ is positive, $(1 + R_{ii}) \geq 0$. Since $\Psi_{ij}$ is a product of two quasi-random numbers, this implies various constraints on the joint properties of $\tilde{R}_{ij}$. Furthermore, these variables are exponentially weakly correlated with $R_{ij}$ of the original ETH ansatz (1),

$$\overline{R_{ij}\tilde{R}_{ji}} = g_{A,\Psi}(e_{ij}^+, \omega_{ij}) e^{-S(e_{ij}^+)/2} , \quad (8)$$

where $g_{A,\Psi}(e_{ij}^+, \omega_{ij})$ is an order one smooth function of its variables, which describes the correlations crucial for non-equilibrium dynamics. The existence of such correlations between the off-diagonal products of $\Psi_{ij}A_{ji}$, can be shown by averaging over the (random) phases of the eigenvectors of $\hat{H}$, see [44].

In particular, for the initial states conforming to Eq.(4), we will assume that a large deviation scaling of the form $\Phi(E) = N\phi(e = E/N)$ applies, such that the following is a *convex function* [45],

$$S(E) - \Phi(E) = N[s(e) - \phi(e)] . \quad (9)$$

Summarising, in the *out-of-equilibrium ETH*, observables and the initial state look like pseudorandom matrices with smooth statistical properties describing correlations or variance of the off-diagonal matrix elements,

$$\overline{|A_{ij}|^2} = e^{-Ns(e^+)} |f_A(e^+, \omega_{ij})|^2, \quad (10a)$$

$$\overline{|\Psi_{ij}|^2} = \frac{e^{-2N\phi(e^+)}}{Z^2} e^{-\frac{\phi''(e^+)}{4N}\omega_{ij}^2}, \quad (10b)$$

$$\overline{\Psi_{ij}A_{ij}} = e^{-Ns(e^+)} \frac{e^{-N\phi(e^+)}}{Z} f_A(e^+, \omega_{ij}) g_{A,\Psi}(e^+, \omega_{ij}), \quad (10c)$$

where we made explicit the dependence on the system size $N$. Eq.(10b) applies to the states conforming to (4) and it is obtained by expanding $E_{i,j} = E^+ \pm \omega_{ij}/2$ around $E^+$ using the assumption of large deviation. A similar term, $e^{-\frac{\phi''(e^+)}{8N}\omega_{ij}^2}$, should appear also in Eq.(10c), however, in the limit of large $N$ this can be neglected because $f_A(e,\omega)$ is expected to decay at large frequencies.

Finally, we note that the product $\Psi_{ij}A_{ij}$ has large fluctuations compared to the average (10c), see [44].

**Consistency checks** – Let us first see how the ansatz (6) and in particular the diagonal ensemble derived from that, satisfy the assumptions (4). The large deviation scaling leads to an ensemble strongly peaked around the characteristic (extensive) energy which maximizes (9) and with sub-extensive fluctuations. In fact, in the large $N$ limit, the energy uncertainty reads:

$$\Delta_{E_0}^2 \equiv \langle\psi|(\hat{H} - E_0)^2|\psi\rangle = \frac{1}{\Phi''(E_0) - S''(E_0)} . \quad (11)$$

Owing to the extensivity in Eq.(9), this implies that $\Delta_E$ is sub-extensive, in particular, for a post-quench state,

$$\Delta_{E_0} = \delta_{e_0}\sqrt{N} ,$$

where $\delta_{e_0}$ is an order-one constant, determined by the shape of the large deviation [46]. We shall now proceed to discuss a set of consistency checks to validate our proposed approach.

*Normalization* – The state normalisation $\text{Tr}\,\Psi = 1$ is ensured by the definition of $Z$. In the large $N$ limit, the "partition function" $Z$ in Eq.(6) reads:

$$Z = \sqrt{2\pi}\Delta_{E_0}e^{S(E_0)-\Phi(E_0)} , \quad (12)$$

with $\Delta_{E_0}$ given by Eq.(11). However one can show a stronger property, namely that $\text{Tr}\,\Psi^2 = 1$. See Eq.(16) below at time zero.

*Projector* – We now discuss an even tighter constraint: the projector identity $\Psi^2 = \Psi$ at the level of individual matrix elements. For our ansatz, this turns out to be true in a statistical way in the thermodynamic limit. In particular, thinking of the matrix elements $\Psi_{ij}$ as products of two random variables we assume the following properties:

$$\tilde{R}_{ik}\tilde{R}_{kj} = \tilde{R}_{ij}(1 + \tilde{R}_{kk}), \quad \text{for } i \neq j \neq k,$$

$$\tilde{R}_{ij}\tilde{R}_{ji} = (1 + \tilde{R}_{ii} + \tilde{R}_{jj} + \tilde{R}_{ii}\tilde{R}_{jj}), \quad \text{for } i \neq j.$$

$$(13)$$

At the leading order in $N$, this implies, see [44],

$$[\Psi^2]_{ij} \simeq \frac{e^{-\frac{1}{2}(\Phi(E_i)+\Phi(E_j))}}{Z}\tilde{R}_{ij} \simeq [\Psi]_{ij} \quad i \neq j, \quad (14a)$$

$$[\Psi^2]_{ii} \simeq \frac{e^{-\Phi(E_i)}}{Z}\left(1 + \tilde{R}_{ii}\right) \simeq [\Psi]_{ii}. \quad (14b)$$

Therefore the ansatz preserves its structure upon multiplication.

**Implications for the dynamics** – We now discuss the main motivation behind our ansatz, designed to describe equilibration dynamics of physical observables.

*Fidelity decay* – First of all, we show that our ansatz is consistent with the expected behavior of the fidelity decay (survival probability) [47–50], defined as

$$|\langle\psi|\psi(t)\rangle|^2 = \left|\sum_i |c_i|^2 e^{-iE_i t}\right|^2 . \qquad (15)$$

By substituting sums with integrals, neglecting spectral correlations, and using the out-of-equilibrium ETH ansatz in Eq.(10b), at the leading order in $N$, one finds

$$|\langle\psi|\psi(t)\rangle|^2 \simeq \frac{1}{2\Delta_{E_0}\sqrt{\pi}}\int d\omega e^{-\frac{1}{2}\frac{1}{2\Delta_{E_0}^2}\omega^2}e^{i\omega t} = e^{-t^2\Delta_{E_0}^2} , \qquad (16)$$

where we have used the definition of the energy variance in Eq.(11), i.e. $\Delta_{E_0} = 1/\sqrt{\Phi''(E_0) - S''(E_0)} = \delta_{e_0}\sqrt{N}$. Thus the large deviation ansatz in Eq.(6) is consistent with the Gaussian decay, controlled by the energy variance of the initial state, in agreement with the literature on global quenches, see e.g. Ref. [48]. Dynamical behavior, different from Eq.(16), e.g. including an exponential decay, is known to arise from the initial states which are different from Eq.(4) [47–49, 51–53], as discussed below.

*Relaxation dynamics* – The primary purpose of this work is the study relaxation dynamics in Eq.(2), namely:

$$\delta A_\Psi(t) = \langle\psi|\hat{A}(t)|\psi\rangle - \langle\hat{A}\rangle_{\text{diag}} = \sum_{i\neq j} c_i c_j^* A_{ij} e^{i(E_i - E_j)t} . \qquad (17)$$

Plugging Eq.(10c) into Eq.(17), the standard ETH manipulations lead to

$$\delta A_\Psi(t) \simeq \int d\omega f_A(e_0,\omega)g_{A,\Psi}(e_0,\omega)e^{\frac{\omega^2}{4N\delta_{e_0}^2}}e^{-i\omega t} . \qquad (18)$$

Hence, the correlation between the initial state and the operator encodes the Fourier transform of the relaxation:

$$\delta\tilde{A}_\Psi(\omega) = f_A(e_0,\omega)g_{A,\Psi}(e_0,\omega) , \qquad (19)$$

where we have neglected the Gaussian frequency term for $N \gg 1$. Thus, the out-of-equilibrium behavior is encoded in this function and will depend, in general, on the correlations between the state and the observable.

The relaxation dynamics (17) share some properties with the (two-time) dynamical correlations at thermal equilibrium for the same observable. One has [4, 5]:

$$C(t) = \frac{1}{2}\langle\{A(t), A(0)\}\rangle_c = \int d\omega\, e^{i\omega t}\cosh\left(\frac{\beta\omega}{2}\right)f_A^2(e_\beta,\omega) , \qquad (20)$$

where $\langle\cdot\rangle = \text{Tr}(e^{-\beta H}\cdot)/\text{Tr}(e^{-\beta H})$ and $e_\beta = \langle H\rangle/N$. Therefore the ETH function $f_A(e,\omega)$ enters both Eqs. (18) and (20) and its properties in the $\omega \to 0$ limit control the long-time behavior. This fact is usually invoked in the literature, see e.g. [2, 5], and our ansatz in Eq. (18) makes it explicit.

Similarly to $f_A^2(\omega)$, which has to decay exponentially at large $\omega$ in $D \geq 2$ (superexponentially in 1D), high-frequency tail of $g_{\Psi,A}(\omega)$ has to be exponentially suppressed for states $\Psi$ associated with local perturbations, see [44].

Let us now comment on some differences between Eq. (18) and (20). The integrand in (20) is positive-definite. As a result, $C(t)$ necessarily decays at early times. On the contrary, the integrand in Eq. (18) is not sign-definite, hence $\delta A_\Psi(t)$ can both increase or decrease throughout relaxation dynamics.

**Numerical results -** We test the predictions above in the case of the one-dimensional Ising model with a tilted field

$$H = \sum_{i=1}^{L} w\sigma_i^x + \sum_{i=1}^{L} h\sigma_i^z + \sum_{i=1}^{L-1} J\sigma_i^z\sigma_{i+1}^z \qquad (21)$$

with $w = \sqrt{5}/2$, $h = (\sqrt{5}+5)/8$ and $J = 1$ and consider different local single or two sites observables,

$$\hat{A} = \sigma_1^x, \ \hat{A} = \sigma_1^z \quad \text{or} \quad \hat{A} = \sigma_1^z\sigma_2^z . \qquad (22)$$

We consider simple out-of-equilibrium initial states, fully polarized states

$$|\psi\rangle = |\downarrow_\alpha\downarrow_\alpha\cdots\downarrow_\alpha\rangle \qquad (23)$$

in the $\alpha = z$ or $\alpha = y$ directions. We impose periodic boundary conditions on the Hamiltonian in Eq.(21) and restrict the analysis to translationally-invariant sector $k = 0$ with positive parity reflection symmetry. As a technical tool, we use the smoothed average of our energy-resolved data as

$$\overline{[f(x)]}_\tau = \frac{\sum_n f(x_n)\delta_\tau(x - x_n)}{\sum_n \delta_\tau(x - x_n)} , \qquad (24)$$

where, $\delta_\tau(x)$ is a smoothed delta functions such that $\lim_{\tau\to\infty}\delta_\tau(x) = \delta(x)$. In the simulations, we chose a Gaussian smoothing $\delta_\tau(x) = e^{-\frac{\tau^2}{2}x^2}/\sqrt{2\pi/\tau^2}$.

First, we establish that the initial states (23) are consistent with the ansatz in Eq.(6). In Fig. 1a we plot the diagonal ensemble for different length sizes $L = 12, 14, 16, 18$, showing that it obeys the large deviation prediction $\overline{\Psi_{ii}} = \frac{e^{-L\phi(E_i/L)}}{Z}$. This is confirmed by the inset, where we plot the scaling of the initial energy $E_0 = e_0 L$ and variance $\Delta_{E_0}/E_0 = \delta_{e_0}/e_0\sqrt{L}$, c.f. Eq.(11). From a fit of the data, we extract the dimensionless values $e_0 = 0.10$, $\delta_{e_0} = 1.12$. In panel (b),

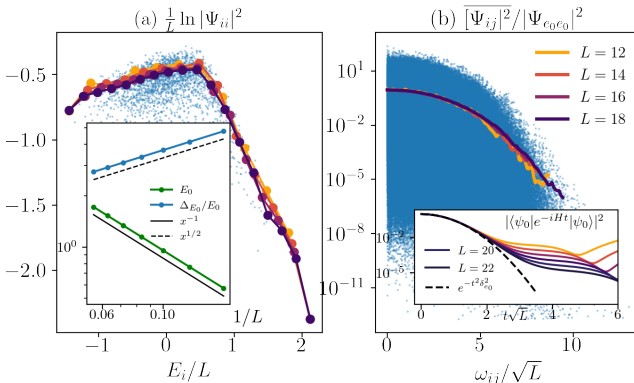

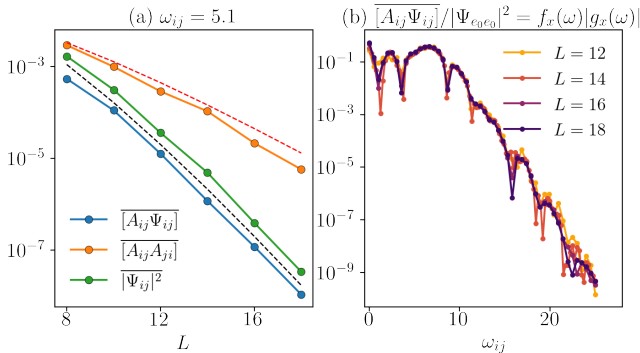

FIG. 1. Out-of-Equilibrium ETH of the fully polarized initial state $|\psi\rangle = |\downarrow_z \ldots \downarrow_z\rangle$. (a) The rescaled diagonal ensemble as a function of the energy density for different system sizes $L = 12, 14, 16, 18$. In the inset, the initial energy $E_0 = \langle\psi|\hat{H}|\psi\rangle$ and the energy fluctuations $(\Delta E_0)^2 = \langle\psi|\hat{H}^2|\psi\rangle - E_0^2$ are plotted as a function of the inverse of the system size $1/L$. In both panels, the blue dots correspond to individual overlaps for $L = 16$. The smoothing parameter is $\tau = 4$. In the inset, the numerical fidelity decay up to $L = 22$ is compared with the prediction in Eq.(16) (dashed) without fitting parameter.

FIG. 2. ETH correlations between the initial state and the observable $A = \hat{\sigma}_1^x$. (a) Scaling with the system size of the ETH predictions in Eqs.(10). The red and black dashed lines indicate $(\dim\mathcal{H})^{-1}$ and $(\dim\mathcal{H})^{-2}$ respectively (b) Smoothed averages $\overline{A_{ij}\Psi_{ij}}$ describing the Fourier transform of the relaxation dynamics increasing system size. Smoothed quantities at energy density $e_0 = 0.10$ with $\tau = 4$.

we study the fluctuations of the out-of-equilibrium ETH functions in the frequency domain [cf. Eq.(10b)]. To address the dependence on $L$, we re-scale $\overline{|\Psi_{ij}|^2}$ by the diagonal matrix elements at energy $e_0$. With this choice, the ansatz in Eq.(10b) is equivalent to

$$\frac{\overline{\Psi_{ij}\Psi_{ji}}}{\left|\Psi_{e^+e^+}\right|^2} \simeq e^{-\frac{\phi''(e^+)}{4L}\omega_{ij}^2}, \quad E^+ = Ne^+ = \frac{E_i + E_j}{2} . \tag{25}$$

In Fig.1b, we fix the energy density to be $e_0$ by resricting the energy indices $i, j$ of $\overline{\Psi_{ij}\Psi_{ji}}$ to $|(E_i + E_j)/2 - E_0| \leq \sqrt{L}\delta_{e_0}$. The figure shows the smoothed average (25) as a function of the energy difference $\omega_{ij} = E_i - E_j$ rescaled by $\sqrt{L}$, for different system sizes. For $L = 16$ we also show individual values without smoothing (blue dots). The plot confirms that this initial state has fluctuations that decay as a Gaussian with a variance $1/\sqrt{L}$, consistent with Eq.(25). In the inset, we also confirm the Gaussian decay of the fidelity upon increasing system size [cf. Eq.(16)]: we plot $e^{-\delta_{e_0}^2 t^2}$ with $\delta_{e_0} = 1.12$ without fitting parameter.

We then proceed to establish the validity of the ansatz for the correlations between the initial state and observable $\hat{A}$ in the energy eigenbasis. In Fig.2, we focus on $|\psi\rangle = |\downarrow_z \ldots \downarrow_z\rangle$ and $\hat{A} = \hat{\sigma}_1^x$. In panel (a), we test the system size dependency of Eqs.(10) at energy density $e_0$ for finite frequency $\omega_{ij} = 5.1$. As predicted by out-of-equilibrium ETH Eq.(10a), the observable off-diagonal matrix elements $\overline{A_{ij}A_{ji}}$ decay as $O(e^{-Ls(e_0)})$, while both $\overline{|\Psi_{ij}|^2}$ and $\overline{A_{ij}\Psi_{ji}}$ decay as $O(e^{-2L})$, cf. Eqs(10b)-(10c). The red and black dashed lines indicate $(\dim\mathcal{H})^{-1}$ and

$(\dim\mathcal{H})^{-2}$ respectively. We checked that the same results hold at zero or for other finite $\omega_{ij}$.
In Fig.2b, we consider

$$\frac{\overline{A_{ij}\Psi_{ij}}}{\overline{\Psi_{e_0e_0}}^2} \simeq f_A(e_0, \omega)g_{A,\Psi}(e_0, \omega_{ij}) , \tag{26}$$

where $\Psi_{e_0e_0}$ is the same as in Eq.(25) and the right-end side follows from Eq.(10c) and (7) [54]. This quantity is of order one, i.e. it remains finite in the thermodynamic limit. Its Fourier transform yields equilibration dynamics, see Eq.(18). Note that we have plotted the absolute value of Eq.(26), since the sign of $g(e, \omega)$ oscillates. This sign change is a characteristic feature of the out-of-equilibrium dynamics, as was emphasized above.

To better understand the behaviour of the function

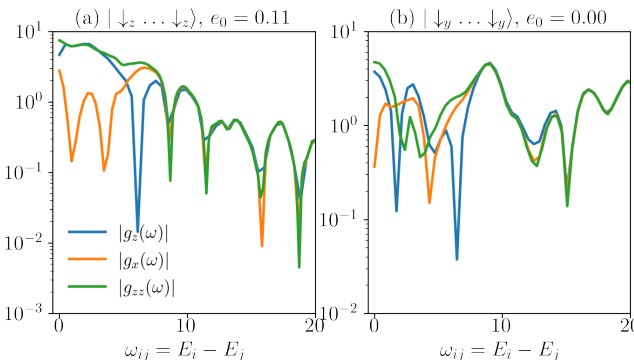

FIG. 3. Absolute value of the correlations between initial state and observable $|g_{A,\Psi}(e_0, \omega)|$ extracted using Eq.(27) for different observables $A = \sigma_1^x$, $\sigma_1^z$ and $\sigma_1^z\sigma_2^z$ as a function of frequency. (a) Results from the initial state $|\downarrow_z \ldots \downarrow_z\rangle$. (b) Results from the initial state $|\downarrow_y \ldots \downarrow_y\rangle$. Here $L = 18$ and $\tau = 4$.

$g(e_0, \omega)$, in Fig.3 we consider:

$$\frac{\overline{A_{ij}\Psi_{ij}}}{\sqrt{\overline{A_{ij}^2}}\sqrt{\overline{|\Psi_{ij}|^2}}\sqrt{\overline{\Psi_{e_0 e_0}}}} \simeq g_{A,\Psi}(e_0, \omega_{ij}) , \qquad (27)$$

to obtain an order one quantity, which encodes the correlations in Eq.(8). The results are shown in Fig.3 for the fully polarized states in Eq.(23) along the directions $\alpha = z, y$ in panels (a) and (b) respectively, or the three different operators in Eq.(22). The plot shows that the $g_{A,\Psi}(e_0, \omega)$ may still decay, albeit slowly, as a function of frequency. The most notable fact is that, for different observables, the smooth functions $g$ have approximately the same behavior at large frequencies, which does not depend on the observable. This seems to indicate that the large frequency behavior and the oscillations in the $g_\psi(e_0, \omega)$ reflect physics *of the initial state*.

**Discussion and conclusions** - In this paper, we have introduced a new ansatz for out-of-equilibrium dynamics, which predicts correlations between the initial state and observables when written in the energy eigenbasis.

Let us remark that our results describe *a wide class of initial states*, including for example products $|a\rangle_A |b\rangle_B$ of energy eigenstates $|a\rangle_A$ and $|b\rangle_B$ of subsystems $A$ and $B$, that have recently motivated studies of the eigenstate correlations [29–34]. For local Hamiltonians, these states have only intensive energy fluctuations $\Delta_E^2 = O(1)$. While obeying Eq.(4), they do not have the form of a large deviation (9) and their survival probability (15) is known to decay exponentially in time [48]. Nevertheless, this class of states is naturally included in our ansatz on the state-observable correlations in Eqs.(10), with a difference in Eq.(10b) which generically reads

$$\overline{|\Psi_{ij}|^2} = e^{-\Phi(E^+ + \omega/2) + \Phi(E^+ - \omega/2)}/Z^2. \qquad (28)$$

These state are discussed in [44], where we verify numerically the general scaling with the system size of Eqs.(10), and comment on the relation with the literature.

Our work opens a series of perspectives. At long times, hydrodynamic modes are expected to play a dominant role in equilibration dynamics [55–57], and it would be valuable to investigate how hydrodynamic description can be incorporated into the non-equilibrium ETH. Additionally, one could explore how the current ETH framework applies to integrable systems that equilibrate to a generalized Gibbs ensemble [11, 42, 58, 59] or in the presence of many-body quantum scars [60–63]. Further questions concern the natural generalization of our ansatz to cross-correlations between different states $c_i^a = \langle E_i | \psi_a \rangle$ for multiple choices of $a$ and $i$ and in the higher-order correlations between states and observables [16, 33, 34].

We thank J. Kurchan for inspiring discussions and collaboration on related topics. We thank A. Rosch and D. Abanin for useful discussions. AD is supported by the NSF grant PHY 2310426. S.P. acknowledges support by the Deutsche Forschungsgemeinschaft (DFG, German Research Foundation) under Germany's Excellence Strategy - Cluster of Excellence Matter and Light for Quantum Computing (ML4Q) EXC 2004/1 -390534769.

* laura.foini@ipht.fr
§ a.dymarsky@uky.edu
† pappalardi@thp.uni-koeln.de

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

$$\Phi''(E_0) = \frac{\partial^2 \Phi(E)}{\partial E^2}\Big|_{E=E_0} = \frac{1}{N}\frac{\partial^2 \phi(e)}{\partial e^2}\Big|_{e=e_0} = \frac{1}{N}\phi''(e_0).$$

Therefore $\delta_{e_0} = 1/\sqrt{\phi''(e_0) - s''(e_0)}$.

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

# Supplemental Material:

In the Supplementary Material, we provide additional analysis and background calculations to support the results in the main text: RMT-based argument for scaling, consistency checks and a bound on high frequency behavior of the out-of-equilibrium ETH ansatz; discussion of bipartite states, etc.

## A SIMPLE SCALING DRAWN FROM RMT

Let us discuss a simple example that exhibits the scaling in Eq.(10). Consider an observable $A = \sum_\alpha \lambda_\alpha |\lambda_\alpha\rangle\langle\lambda_\alpha|$ and as an initial state we will take an eigenvector of such observable $|\psi\rangle = |\lambda_\psi\rangle$. Let us suppose that afterward, the state evolves under a $\mathcal{D}\times\mathcal{D}$ Hamiltonian that is drawn from a rotationally invariant ensemble, i.e. $P(H) = P(U^{-1}HU)$ where $U$ is arbitrary orthogonal (or unitary) matrix, for instance a GOE or GUE ensemble. With this choice, the Hamiltonian eigenvectors $|E_j\rangle$, in the basis of the observable, i.e. $\langle E_j|\lambda_\alpha\rangle$, are represented by random orthogonal or unitary matrices. The properties of the matrix elements of a given observable $A$ in such random basis have been discussed in [64]. In the large $\mathcal{D}$ limit assuming to initialise the dynamics in $|\psi\rangle = |\lambda_\psi\rangle$, some eigenvector of the observable $A$, the properties of this toy "out-of-equilibrium" ETH can be easily derived

$$\overline{|A_{ij}|^2} = \frac{1}{\mathcal{D}}\left(\langle A^2\rangle - \langle A\rangle^2\right) \tag{S1a}$$

$$\overline{|\Psi_{ij}|^2} = \frac{1}{\mathcal{D}^2} \qquad \text{for } i \neq j \tag{S1b}$$

$$\overline{\Psi_{ij}A_{ij}} = \frac{1}{\mathcal{D}^2}\left[\lambda_\psi - \langle A\rangle\right] \tag{S1c}$$

where $\langle\bullet\rangle = \frac{1}{\mathcal{D}}\text{Tr}(\bullet)$. This is a particularly simple example of the ansatz (10) discussed in the main text. In term of normalized fluctuations this means $\overline{R_{ij}\tilde{R}_{ij}} \simeq \mathcal{D}^{-1/2}$, as in (8). This examples illustrates the difference in scaling between $\overline{|A_{ij}|^2}$ and $\overline{|\Psi_{ij}|^2}$. The first quantity has rank $\mathcal{D}$, leading to $\sum_{ij}\overline{|A_{ij}|^2} = O(\mathcal{D})$, while the second has rank one, $\sum_{ij}\overline{|\Psi_{ij}|^2} = O(1)$. Similarly $\sum_{ij}\overline{\Psi_{ij}A_{ij}} = O(1)$, which is consistent with the scaling above.

## A CONSTRAINT ON THE ANSATZ

Let us justify Eqs. (13). As we stressed several times, contrary to the standard ETH ansatz for observables, the matrix that we have are chracterising has rank 1. In particular, calling $z_i = \frac{1}{\sqrt{Z}}e^{-\frac{1}{2}\Phi(E_i)}$ and following the notation of the main text we have:

$$1 + \tilde{R}_{ii} = \frac{|c_i|^2}{z_i^2}$$
$$\tilde{R}_{ij} = \frac{c_i c_j^*}{z_i z_j} \qquad \text{for } i \neq j \tag{S2}$$

Taking products:

$$\tilde{R}_{ik}\tilde{R}_{kj} = \frac{c_i|c_k|^2 c_j^*}{z_i|z_k|^2 z_j} = \tilde{R}_{ij}(1 + \tilde{R}_{kk}) \qquad \text{for } i \neq j \neq k \tag{S3}$$

and similarly

$$\tilde{R}_{ij}\tilde{R}_{ji} = \frac{|c_i|^2|c_j|^2}{|z_i|^2|z_j|^2} = (1 + \tilde{R}_{ii})(1 + \tilde{R}_{jj}) \qquad \text{for } i \neq j \tag{S4}$$

Let us now see how these constraints imply that $\Psi$ is a projector by proving Eqs. (14).
We start by evaluating the off-diagonal with $i \neq j$:

$$[\Psi^2]_{ij} = \sum_k \Psi_{ik}\Psi_{kj} = \Psi_{ii}\Psi_{ij} + \Psi_{ij}\Psi_{jj} + \sum_{k:k\neq i\neq j}\Psi_{ik}\Psi_{kj}$$
$$= \frac{1}{Z^2}e^{-\frac{1}{2}(\Phi(E_i)+\Phi(E_j))}\left[e^{-\Phi(E_i)}\tilde{R}_{ii}\tilde{R}_{ij} + e^{-\Phi(E_j)}\tilde{R}_{jj}\tilde{R}_{ij}\right] + \frac{1}{Z}e^{-\frac{1}{2}(\Phi(E_i)+\Phi(E_j))}\sum_{k:k\neq i\neq j}\frac{e^{-\Phi(E_k)}}{Z}\tilde{R}_{ik}\tilde{R}_{kj} \tag{S5}$$

where from the first to the second line we have inserted the ansatz (10) in the individual matrix elements. The first term is subleading $O(e^{-2N})$, while in the second term, we can substitute the ansatz of Eq.(S3) and obtain

$$[\Psi^2]_{ij} \simeq \frac{1}{Z} e^{-\frac{1}{2}(\Phi(E_i) + \Phi(E_j))} \sum_{k:k \neq i \neq j} \frac{e^{-\Phi(E_k)}}{Z} \tilde{R}_{ij}(1 + \tilde{R}_{kk}) = \frac{1}{Z} e^{-\frac{1}{2}(\Phi(E_i) + \Phi(E_j))} \tilde{R}_{ij} \sum_{k:k \neq i \neq j} \Psi_{kk}$$

$$= \frac{1}{Z} e^{-\frac{1}{2}(\Phi(E_i) + \Phi(E_j))} \tilde{R}_{ij} = \Psi_{ij} ,$$

(S6)

where we used $\sum_{k:k \neq i \neq j} \Psi_{kk} \simeq \sum_k \Psi_{kk} = \langle \psi | \psi \rangle = 1$ which shows the first equation in (14). Similar manipulations can be done on the diagonal elements, leading to

$$[\Psi^2]_{ii} = \sum_k \Psi_{ik} \Psi_{ki} = \Psi_{ii} \Psi_{ii} + \sum_{k:k \neq i} \Psi_{ik} \Psi_{ki}$$

$$= \frac{e^{-2\Phi(E_i)}}{Z^2}(1 + \tilde{R}_{ii})^2 + \frac{1}{Z} e^{-\Phi(E_i)} \sum_{k:k \neq i} \frac{e^{-\Phi(E_k)}}{Z} \tilde{R}_{ik} \tilde{R}_{ki}$$

$$\simeq \frac{1}{Z} e^{-\Phi(E_i)}(1 + \tilde{R}_{ii}) \sum_{k:k \neq i} \frac{e^{-\Phi(E_k)}}{Z}(1 + \tilde{R}_{kk}) = \Psi_{ii} \sum_{k \neq i} \Psi_{kk} = \Psi_{ii} ,$$

(S7)

where, from the second to the third line we have used the fact that the first term is subleading and the ansatz in Eq.(S4).

## FLUCTUATIONS OF STATE-OBSERVABLE CORRELATIONS

At the level of single matrix elements, the product of the initial state and the observable has large fluctuations in the system size. In fact

$$\Psi_{ij} A_{ij} \simeq \overline{\Psi_{ij} A_{ji}} + \sqrt{\overline{|\Psi_{ij}|^2}} \sqrt{\overline{|A_{ij}|^2}} \xi_{ij}$$

(S8)

with $\xi_{ij}$ some random variable with average zero and fluctuations order one. Here, the amplitude of the fluctuations is larger than the average:

$$\sqrt{\overline{|\Psi_{ij}|^2 |A_{ij}|^2}} \gg \overline{\Psi_{ij} A_{ji}} ,$$

since $\sqrt{\overline{|\Psi_{ij}|^2 |A_{ij}|^2}} \simeq e^{-3S/2}$ and $\overline{\Psi_{ij} A_{ji}} \sim e^{-2S}$. However, when computing physical observables, one has to sum over many indices, and, due to the presence of randomness, the fluctuations become negligible. This is analogous to what happens to high-order products of matrix elements in standard ETH, which also possess large fluctuations [15]. These fluctuations contribute, at least for finite systems sizes, to $\langle A(t) \rangle \langle A(-t) \rangle$, A detailed understanding of their influence on the dynamics is left to future work.

## A BOUND ON HIGH FREQUENCY TAIL OF $g(\omega)$

As a starting point we introduce

$$Z(\beta) = \langle \psi | e^{-\beta H} | \psi \rangle,$$

(S9)

generalizing $Z(0)$ defined in (6).

To constrain $g_{\psi,A}$ we use the approach similar to one used [65], which bounds on high-frequency tail of $f_A$,

$$|f_A(\tilde{e}, \omega)| \leq O\left(e^{-(\tilde{\beta}/4 + \beta^*)\omega}\right), \qquad \omega \to \infty.$$

(S10)

where $\beta^*$ is an $O(1)$ constant defined by local model parameters. Here $O(\dots)$ means that possible pre-exponential $\omega$-dependent factors are ignored. Finally, temperature $\tilde{\beta}$ is associated with energy density $\tilde{e}$, $S'(N\tilde{e}) = \tilde{\beta}$. We now consider

$$C_\Psi^\beta(t) \equiv \frac{\langle \psi | e^{-\beta H} A(t) | \psi \rangle}{Z(\beta)}.$$

(S11)

We now use the following inequality

$$|\langle\psi|A|\psi'\rangle| \leq |A||\psi||\psi'|, \tag{S12}$$

where $|A|$ is an infinity norm of the operator $A$, meaning the largest (by absolute value) eigenvalue when $A$ is hermitian, or largest singular value when $A$ is not hermitian. Taking $|\psi\rangle = |\psi\rangle$ and $\langle\psi'| = \langle\psi| e^{-\beta H}$ we arrive at

$$\left|\int d\omega\, f_A(\tilde{e},\omega)g_{\psi,A}(\tilde{e},\omega)e^{\omega(it-\beta/2)}\right| \leq |A(t)|\frac{Z^{1/2}(2\beta)Z^{1/2}(0)}{Z(\beta)}. \tag{S13}$$

Here $\tilde{e}$ is the energy density where the main contribution to the integral in (S9) comes from, $\tilde{e} = -\partial/\partial\beta \ln Z(\beta)/N$. We can now redefine $t \to t - i\beta/2$,

$$\left|\int d\omega\, f_A(\tilde{e},\omega)g_{\psi,A}(\tilde{e},\omega)e^{i\omega t}\right| \leq |A(t-i\beta/2)|\frac{Z^{1/2}(2\beta)Z^{1/2}(0)}{Z(\beta)}. \tag{S14}$$

The LHS is an even function of $t$, while $|A(t)|$ is analytic within the strip $|\text{Im}(t)| \leq \beta^*$ [65]. Thus the RHS of (S14) is analytic inside the strip $-\beta^* + \beta/2 \leq \text{Im}(t) \leq \beta^* + \beta/2$. Because the LHS is even, it has to be analytic inside a wider strip $-\beta^* - \beta/2 \leq \text{Im}(t) \leq \beta^* + \beta/2$. For the integral over $\omega$ to converge, taking into account the bound (S10) we find

$$|g_{\psi,A}(\tilde{e},\omega)| \leq O\left(e^{-(\beta/2-\tilde{\beta}/4)\omega}\right)\frac{Z^{1/2}(2\beta)Z^{1/2}(0)}{Z(\beta)}, \qquad \omega \to \infty. \tag{S15}$$

Here $\beta$ is a free parameter, it determines the "saddle point" (mean energy density) $\tilde{e}(\beta)$, where the integral in (S9) is saturated, which in turn defines $\tilde{\beta}$. Parameter $\beta^*$, which characterizes the model does not appear in (S15). When $\beta = 0$, mean energy density $\tilde{e} = e_0$, and $\tilde{\beta}$ is the effective temperature of state $\psi$.

The logic behind free parameter $\beta$ appearing in bound (S15) is exactly as in [65], this is a parameter to optimize over, to find the best possible bound. For the large deviation states (9), the factor $\frac{Z^{1/2}(2\beta)}{Z(\beta)}$ grows extensively, $\ln(\frac{Z^{1/2}(2\beta)}{Z(\beta)}) \sim N\, O(\beta^2)$, not leading to a meaningful bound in the thermodynamic limit.

For the states with energy of order one, e.g. discussed in the next section, effective energy density $\tilde{e} = e_0$ is $\beta$-independent, $\tilde{\beta} = \beta_0$, at least so for the parameter $\beta$ smaller than certain value $\beta < \lambda$, see (S18). In this regime $Z^{1/2}(2\beta)Z^{1/2}(0)/Z(\beta)$ is of order one, and $g_{\psi,A}(\tilde{e},\omega)$ for large $\omega$ decays exponentially, bounded by $e^{-(\lambda/2-\beta_0/4)\omega}$, provided $\beta_0 \leq 2\lambda$.

## BIPARTITE ENERGY EIGENSTATES

Consider a tensor product Hilbert space $\mathcal{H} = \mathcal{H}_A \otimes \mathcal{H}_B$ and a Hamiltonian $\hat{H} = \hat{H}_A + \hat{H}_B + \hat{H}_{AB}$, describing an interaction of a (sub)system $A$ with a "bath" $B$. A particular example to keep in mind is a spin-chain split into two parts, interacting through a local term $H_{AB}$. We start with a pair $|a\rangle$, $|b\rangle$ of the eigenstates of $\hat{H}_A$, $\hat{H}_B$ respectively and consider a state $|\psi\rangle = |ab\rangle$. Decomposition of this state in the eigenbasis $|E_i\rangle$ of $\hat{H}$ defines the projector (6),

$$\Psi_{ij}^{(ab)} = \langle E_i|ab\rangle\langle ab|E_i\rangle = c_i^{(ab)}c_j^{(ab)*}. \tag{S16}$$

These initial states have been the focus of a great attention lately [27, 29–34, 66], , especially in relation to bipartite entanglement. The statistical properties of (S16) can be formulated in terms of the so-called *Ergodic Bipartition* (EB) ansatz [34], which postulates that on average

$$\overline{\Psi_{ii}^{(ab)}} = \overline{|c_i^{(ab)}|^2} = e^{-S(E_a+E_b)}F(E_i - E_a - E_b), \tag{S17}$$

where $F(E_i - E_a - E_b)$ is a narrowly-peaked function around $E_i \simeq E_a + E_b$. More accurately instead of $E_a + E_b$ in the expression above one should use mean energy $E_0$ of $|ab\rangle$, as we do below. More detailed properties of $F(x)$ for 1D systems with local interactions, the Lorentzian shape at small $x$ and exponential suppression at large $x$,

$$F(x) \propto \begin{cases} (x^2 + \Delta^2)^{-1}, & x \ll \sigma, \\ e^{-|x|\lambda}, & x \gg \sigma, \end{cases} \tag{S18}$$

where $\sigma, \Delta, \lambda$ are model-dependent local (finite in the thermodynamic limit) parameters, were outlined in [32].

From the definition (S16) it is clear the ergodic bipartition ansatz (S17) is the diagonal part of the out-of-equilibrium ETH ansatz (6) applied to a particular initial states. We now discuss how our approach encompasses the properties of (S17). Starting from (7), using saddle point approximation we find,

$$\overline{\Psi_{ii}} = \frac{e^{-\Phi(E_i)}}{Z} \approx e^{-S(E_0)} \frac{e^{-\Phi(E_i)+\Phi(E_0)}}{\sqrt{2\pi}\Delta_{E_0}}, \tag{S19}$$

where $E_0$ is the mean energy of state $|ab\rangle$, $E_0 = E_a + E_b + \Delta_{ab}$. Here $\Delta_{ab} = \langle ab|\hat{H}_{AB}|ab\rangle$ and

$$\Delta_{E_0}^2 = \langle ab|\hat{H}_{AB}^2|ab\rangle - (\Delta_{ab})^2 = \mathcal{O}(1)$$

are of order one, i.e. remain finite in the thermodynamic limit, while $E_0$ and $E_a + E_b$ are extensive. From here follows

$$F(E_i - E_0) \approx \frac{e^{-\Phi(E_i)+\Phi(E_0)}}{\sqrt{2\pi}\Delta_{E_0}} \tag{S20}$$

which is sharply peaked around mean energy $E_i \approx E_0$ with the variance of order one.

Integrating (S19) over $dE_b$ with the weight $e^{S_B(E_b)}$ and the constraint $E_0 \approx E_a + E_b$ readily gives diagonal approximation to reduced density matrix [27]

$$\text{Tr}_B(|E_i\rangle\langle E_i|) \approx \int dE_a\, e^{S_A(E_a)} e^{-S(E_i)+S_B(E_i-E_a)} |a\rangle\langle a|, \tag{S21}$$

from where von Neumann and Renyi entropy follow via saddle point approximation. Justification of the diagonal approximation to evaluate entropy was recently addressed in [34] by considering the statistical properties of

$$c_i^{(ab)} c_i^{(a'b)*} c_i^{(a'b')} c_i^{(ab')*}, \tag{S22}$$

which can be understood extending the ansatz to four different states $|\psi_1\rangle = |ab\rangle$, $|\psi_2\rangle = |a'b\rangle$, $|\psi_3\rangle = |a'b'\rangle$, $|\psi_4\rangle = |ab'\rangle$, with $i_1 = i_2 = i_3 = i_4$.

To illustrate the behavior of $\Phi(E_i)$ we consider, c.f. (S18),

$$\ln\overline{\Psi_{ii}} \simeq -S(E_0) - \frac{1}{2}\ln(2\pi\Delta E^2) - \frac{(E_i-E_0)^2}{2\Delta_{E_0}^2} + \dots \tag{S23}$$

and note that for $E - E_0$ of order one, higher-order corrections to (S23) are also of order one and can not be neglected [32]. It corresponds to the first two terms of the Taylor expansion of (S18) in $(E_i - E_0)^2$. We plot $\phi(E_i) \equiv \ln\overline{\Psi_{ii}} + S(E_0) + \frac{1}{2}\ln(2\pi\Delta E^2)$ as a function of $E_i - E_0$ numerically in the left panel of Fig. S1 for the tilted field Ising model (21) of size $L = 2L' + 1$ with the parameters $J = -1, w = 1.05, h = 0.4$. The eigenstates $|a\rangle, |b\rangle$ are chosen to be the ground state and the most excited states of the subsystems of size $L'$ and $L' + 1$ correspondingly. With this choice of $\psi_0$ the total energy $E_0$ is very close to the middle of the spectrum. To obtain $\overline{\Psi_{ii}}$ we use smoothed average (24) with $\tau = 2$. The entropy

$$S(E) = \ln\Omega(E_0), \qquad \Omega(E) = \frac{\kappa\, L!}{(L/2 - \kappa E)!(L/2 + \kappa E)!} \tag{S24}$$

is evaluated using binomial analytic approximation for the density of states of the titled filed Ising model, see Appendix A of [27], with $\kappa = \frac{1}{2}\sqrt{J^2 + w^2 + h^2 - 1/L}$. Mean energy $E_0$ and energy variance $\Delta_{E0}^2$ are evaluated numerically for each $L$. The plot in S1 shows good collapse of $\phi(E_i - E_0)$ for different values of $L$, and is well described by $\frac{(E_i-E_0)^2}{2\Delta_{E0}^2}$ for small $|E_i - E_0|$.

The out-of-equilibrium ETH anzats predicts the off-diagonal matrix elements

$$\varphi(\omega) = \ln\overline{\Psi_{ij}^2} - 2\ln\overline{\Psi_{ee}} \simeq -\frac{\omega^2}{4\Delta_{E_0}^2}, \quad |\omega| \lesssim \Delta_{E_0}, \tag{S25}$$

plotted in the right panel of Fig. S1. Here $\overline{\Psi_{ee}}$ denotes the average $\overline{\Psi_{ii}}$ over a narrow shell around $E_i = E_0$ of size $\Delta_{E_0}$. Similarly, $\Psi_{ij}^2$ in (S25) is averaged only over pairs $E_i, E_j$ satisfying $|(E_i + E_j)/2 - E_0| \leq \Delta_{E_0}$.

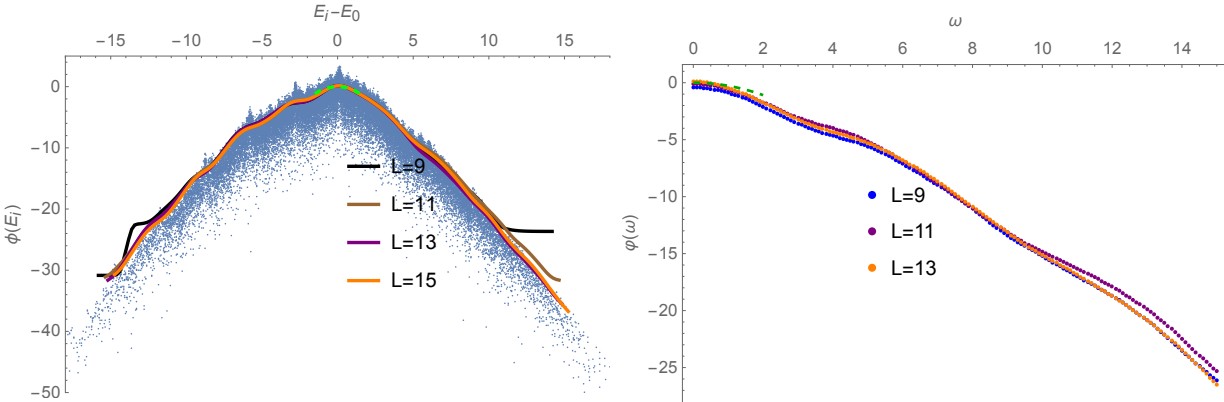

FIG. S1. Left panel: averaged diagonal matrix elements (S19), with the entropy subtracted $\phi(E_i) \equiv \ln \overline{\Psi_{ii}} + S(E_0) + \frac{1}{2}\ln(2\pi\Delta E^2)$, for different system sizes, superimposed with $-(E_i - E_0)^2/(2\Delta_{E_0}^2)$ with the value of $\Delta_{E_0}^2$ for $L = 15$ (green dashed line). Blue points show raw data (un-averaged value of $\ln\Psi_{ii}$, with the entropy subtracted) for $L = 15$. Right panel: average of the off-diagonal matrix elements (S25) for different system sizes, superimposed with $-\omega^2/(4\Delta_{E_0}^2)$ with the value of $\Delta_{E_0}^2$ for $L = 13$ (green dashed line).

Next, we study cross-correlations of $\psi$ with different operators $A$, sitting at the L(eft) edge, R(ight) edge, or the M(iddle) site of the chain,

$$A = \sigma_L^{z,x} = \sigma_{site=1}^{z,x} , \quad A = \sigma_M^{z,x} = \sigma_{site=L'+1}^{z,x} , \quad A = \sigma_R^{z,x} = \sigma_{site=L}^{z,x} . \tag{S26}$$

We plot $f_A(\omega)g_{\psi,A}(\omega)$ for different $A$ in Fig. S2. In the second row of Fig. S2, we display the scaling with the system size of Eqs.(10), finding a good agreement with the predictions at zero and finite frequency ($\omega = 2.1$ shown in the Figure).

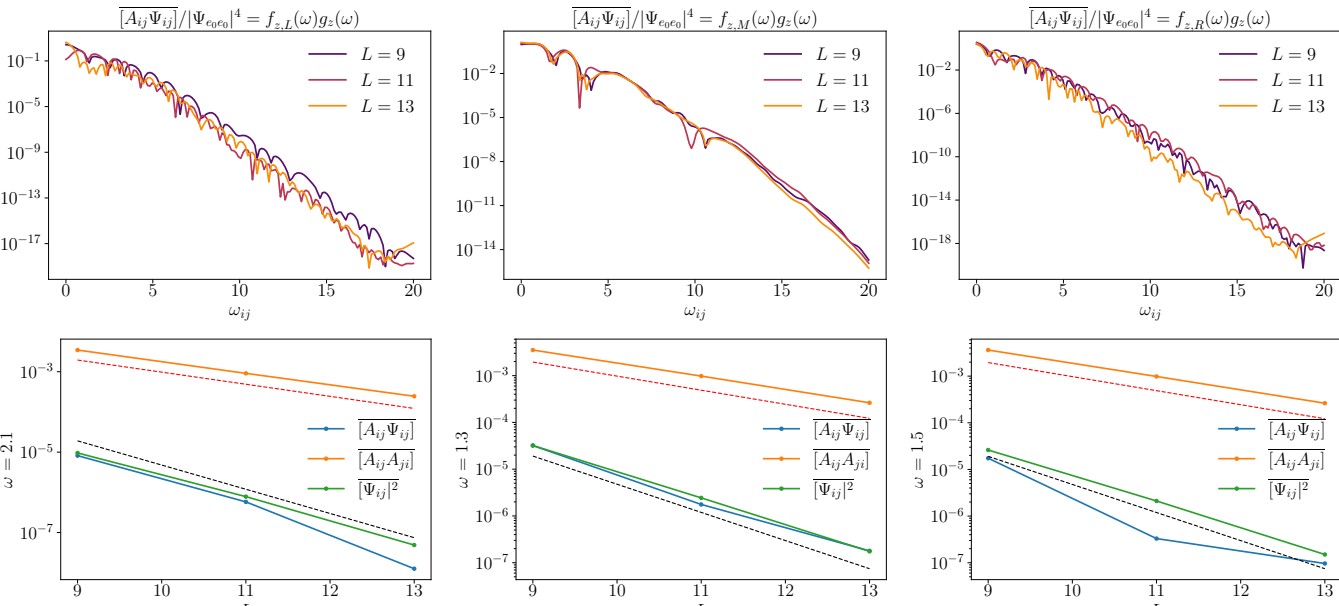

FIG. S2. First row: from left to right, plots of $f_A(\omega)g_{\psi,A}(\omega)$ for $A = \sigma_L^z, \sigma_M^z, \sigma_R^z$ correspondingly. Second row: scaling of Eqs.(10) contrasted with $(\dim\mathcal{H})^{-1}$ (dashed red line) and $(\dim\mathcal{H})^{-2}$ (dashed black line).

Finally, in Fig. S3 we illustrate approximate independence of $g_{\psi,A}(\omega)$ at high frequencies, on the choice of the observable $A$.

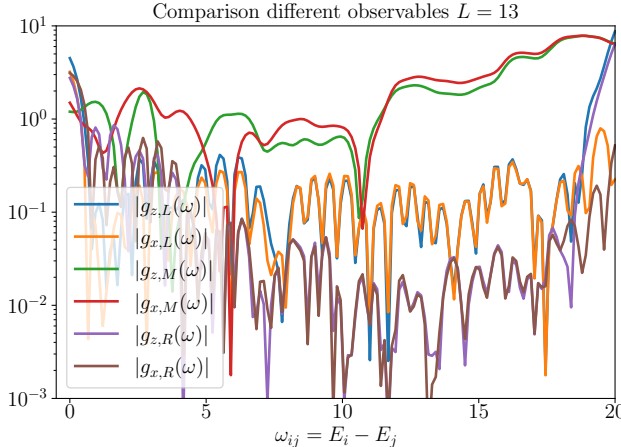

FIG. S3. In the absence of translational invariance, the function $g$ is independent of the direction of the observable, but not on the site.