# Peer review of "Out-of-equilibrium Eigenstate Thermalization Hypothesis"

_SciPost Physics_

## Round 3 · Referee Report · Anonymous (Referee 1) · 2024-11-11

Strengths

The paper is certainly interesting and perhaps even overdue as since early days of ETH there were many discussions of relation between ETH and relaxation of observables. It was also realized that the key are the correlations between the matrix elements of observables and the overlaps of the initial state with the eigenstates but to my knowledge this is the first serious work in this direction.

I think the paper is well written and well organized. In addition it contains various numerical and analytical tests of consistency of results so there is no question in my mind that it fully deserves publication.

Report

Below I want to list some questions and suggestions, which are optional, and the authors might (or might not) want to address them.

  1. The authors effectively study cross-correlations between the matrix elements of the i) projector operator to the initial state $P=|\Psi\rangle\langle \Psi|$, which, as they note, is the rank one operator and has special properties, and ii) the matrix elements of some observable $A$. I think it would be nice first to formulate similar cross-correlation function between two observables say $A$ and $B$ and define the analogue of the function $f(\omega) g^\ast (\omega)$. When $A=B$ this would be $|f(\omega)|^2$, which is studied a lot in the literature, but generally $f\neq g$. After formulating this cross-correlation for arbitrary operators it would be easier to understand the subtleties peculiar to $ B=P$.

  2. From Eq. (5) it follows that $|\Psi_{ij}|^2=\Psi_{ii}\Psi_{jj}$ , which seems to suggest that there are roughly $D^2$ independent phases in $\Psi_{ij}$ but only $D$ independent amplitudes. I wonder if this constraint plays any role in defining $\tilde R_{ij}$. Perhaps this is what allows the authors to derive Eq. (13), but maybe the authors could say in words what is important. I clearly can extend Eq. (13) to add products with more terms and more constraints on $\tilde R$. Do they follow automatically from (13) or are there more constraints?

  3. Related to the previous question. Can I use this machinery to study objects like $\langle \psi | A(t) A(t') |\psi\rangle$ or one must introduce more functions like in higher odder ETH the authors developed. Maybe a general comment of how one can use the formalism for practical calculations. By the way, I am not sure that "out of equilibrium ETH" is a good title. ETH is still equilibrium to my taste as the authors use equilibrium eigenstates, just as I wrote, the important point is that one of the operators is the projector operator. If we e.g. use a projector to a mixed state, for example a pure polarized state for 10 sites in the middle and the identity matrix or some thermal state for the rest. For which length of the partial projector the suggested ETH will become equilibrium then? 4, I wonder if the authors can comment on how their formalism connects with the Kubo linear response when the state $|\Psi\rangle$ is a perturbed eigenstate of the Hamiltonian with say the operator $\epsilon A$ or more generally $\epsilon B$. In this case $\langle \Psi| A(t) |\Psi\rangle$ should reduce to a standard integral of a two-point function and hence be expressed through the standard ETH. This question connects to my question 1.

  4. In Fig. 2 there seem to be a good convergence to the thermodynamic limit at large frequencies as perhaps expected. Can the authors make some claims about say short time relaxation. Does this convergence differ qualitatively from convergence of the cross-correlation (or the spectral function) of normal observables? Does the new ETH formalism allow them to reduce finite size effects?

  5. The easiest one :). There is a typo right below Eq. (28).

Recommendation

Publish (surpasses expectations and criteria for this Journal; among top 10%)

---

## Round 3 · Referee Report · Anonymous (Referee 2) · 2024-11-13

Report

In this paper an extension of the eigenstate thermalisation hypothesis is formulated. The extension makes it possible to discuss relaxation after a quantum quench in chaotic many-body quantum systems. It does so by formulating an Ansatz for matrix elements (in the basis of eigenstates of the Hamiltonian for the system) of a projector onto an initial state, and for matrix elements of an observable in the same basis. The Ansatz is statistical, and correlations between the two sets of matrix elements are a central part of it. Various statistical checks of the Ansatz are given and it is tested against numerics.

The work constitutes an important development in the area of chaotic many-body dynamics and is likely to become a standard reference in the field. For these reasons, it easily satisfies SciPost's second acceptance criterion, of "opening a new pathway in an existing or a new research direction, with clear potential for multi-pronged follow-up work".

The paper is generally clearly written and will be easy to follow for others working in the field.

I recommend publication after the authors have considered the following points:

[1] There are references in the text to [44] Supplemental Material. Is this in fact the appendices? If so, please correct the reference. I could not find any other material.

[2] The caption for Fig 1 refers to panel (a) but not panel (b). Although panel (b) is mentioned in the main text, I think the figure caption should be extended.

[3] Can the data in Fig 1(b) be used to test the probabilty distribution of $\tilde{R}_{ij}$? If so, I suggest that this would be a useful addition to the paper.

[4] I found the notation of Eq 37 hard to decipher. It is written that $Z(\beta)$ generalises $Z(0)$ defined in (6) but: (i) the notation $Z(0)$ does not appear in (6) and (ii) if I set $\beta=0$ in (37) I appear to get $Z(0)=1$. I suggest this should be clarified.

[5] I noticed two typos:
(a) in the text above (8) I think $R_{ij}$ should be $\tilde{R}_{ij}$;
(b) in (18) the sign in the exponential is probably wrong.

Recommendation

Publish (surpasses expectations and criteria for this Journal; among top 10%)

---

## Round 5 · Referee Report · Anonymous (Referee 2) · 2025-3-28

Report

For a general assessment, see my report on the first version. In the second version the authors have addressed in a satisfactory way the points I made in my original report.

Recommendation

Publish (surpasses expectations and criteria for this Journal; among top 10%)

---

## Round 5 · Author Response

We thank both referees for the appreciation of our work and the constructive reports which allowed
us to improve our manuscript. Below you will find the answers to their comments.

---

## Round 5 · List of Changes

Referee 1

1) We thank the referee for this point.
For the correlator $\langle A(t)B\rangle$ similar considerations lead to $F_{AB}^{(n)}(\omega)=f_A(\omega)f_{B}(-\omega)g_{AB}(\omega)$ with $g_{AB}(\omega_{ij}) = \overline{R_{ij}^A R^B_{ji}}$. This was mentioned, in different terms, already in Ref.[Phys. Rev. Lett. 125, 050603 (2020)] and Ref.[Phys. Rev. Lett. 129, 170603 (2022)]. We introduced a footnote on page 3 to highlight this point. Since our main focus is the study of post-quenched dynamics, we have decided not to discuss these cross-correlations in detail.

2) The Referee is right, $\Psi_{ij}$ being of rank-1 implies additional higher-order relations, for instance $\tilde{R}_{ij}\tilde{R}_{jk} \tilde{R}_{ki} = (1+\tilde{R}_{ii})(1+\tilde{R}_{jj})(1+\tilde{R}_{kk})$ with $i\neq j\neq k$ and so on. It seems that they all descend from (14a) and (14b).
At order $3$, using first (14a) and then (14b), we find
\begin{equation}
\tilde{R}_{ij}\tilde{R}_{jk} \tilde{R}_{ki} = \tilde{R}_{ik} (1+ \tilde{R}_{jj} )\tilde{R}_{ki} = (1+\tilde{R}_{ii})(1+\tilde{R}_{jj})(1+\tilde{R}_{kk}).
\end{equation}
Similarly at order $4$, using twice (14a) and then (14b).
\begin{equation}
\tilde{R}_{ij}\tilde{R}_{jk} \tilde{R}_{kl}\tilde{R}_{li} = \tilde{R}_{ik} (1+ \tilde{R}_{jj} )\tilde{R}_{ki}(1+\tilde{R}_{ll} ) = (1+\tilde{R}_{ii})(1+\tilde{R}_{jj})(1+\tilde{R}_{kk})(1+\tilde{R}_{ll})
\end{equation}
We expanded Appendix A.2 to mention these constraints there.

3) We thank the referee for the remark on multi-time correlation functions, which we believe is very interesting.
To properly capture such correlation functions, our ansatz has to be extended. In the revised version, we discuss this in section 2.4. Regarding the title, indeed our ansatz is useful beyond out-of-equilibrium context, but we wanted to emphasize the dynamical aspect, not captured by conventional ETH.

4) Let us consider a state which is a small perturbation of an eigenstate, $| \psi \rangle= e^{i \epsilon B}|E_n\rangle$. By expanding for small $\epsilon$ one gets:
$$|\psi\rangle = \frac{1}{\sqrt{1+\epsilon^2 \langle E_n| B^2|E_n\rangle}} \left( |E_n\rangle + i \epsilon B|E_n\rangle \right)$$
so that $\langle \psi| A(t) | \psi\rangle \simeq \langle E_n | A |E_n\rangle + i \epsilon \langle E_n| [ A(t) , B ] |E_n\rangle$.
We readily find
\begin{equation}
\Psi_{ij} \simeq \delta_{i,n}\delta_{j,n}- i\epsilon [B_{in} \delta_{j,n} -B_{nj} \delta_{i,n} ] = \delta_{i,n}\delta_{j,n}- i\epsilon e^{- S(E_n)/2} [f_B(\omega_{in}) R_{in}^B \delta_{j,n} -f_B(\omega_{jn}) R_{nj}^B \delta_{i,n}].
\end{equation}
Tensor $\Psi_{ij}$ for this state doesn't have a smooth diagonal component, it is thus beyond the scope of our ansatz.
This is because the perturbation is applied to an eigenstate. If instead the parturbation is applied to the type of states that we consider, the ansatz is stable. We comment on this in section 2.4.
Generalization of ansatz to include the states proposed by the referee would be an interesting task for the future.

5) From the numerical results we have, it seems that the ``out-of-equilibrium ETH'' ansatz has the same finite-size effects as the standard one. In fact, the convergence with the system size $N$ in Fig.~3 of the manuscript is very similar to the one we obtain for the matrix elements $\overline{A_{ij}A_{ji}}$. We wanted to include a figure to show better this but it is not possible in the text version that we use here.

6) We thank the referee for the careful reading and have corrected the typo.

Referee 2

1) We thank the referee for careful reading; we have removed the reference to the supplementary material, which was unnecessary.

2) We have extended the caption of Fig.1 to add a description of panel (b).

3) To first approximation, the probability distribution of the off-diagonal matrix elements follows that of the product of two uncorrelated Gaussian random variables, resulting in a distribution described by the modified Bessel function of the second kind. We have added a discussion of this point in section 2.4 of the manuscript, and show corresponding numerical results in Fig.~2 in the revised version.

4) We thank the referee for pointing out a set of confusing notations. We have address that in the revised version.

5) We thank the referee for the careful reading. We have corrected the typos.

---

## Editorial Decision

accepted_in_target_journal